# Effect of Discharge Energy on Micro-Arc Oxidation Coating of Zirconium Alloy

**DOI:** 10.3390/ma17133166

**Published:** 2024-06-27

**Authors:** Wei Wang, Kai Lv, Zhaoxin Du, Weidong Chen, Zhi Pang

**Affiliations:** 1School of Materials Science and Engineering, Inner Mongolia University of Technology, Hohhot 010051, China; 18272971158@163.com (W.W.); duzhaoxin@imut.edu.cn (Z.D.); weidongch@163.com (W.C.); pz369059026@163.com (Z.P.); 2Engineering Research Center of Development and Processing Protection of Advanced Light Metals, Ministry of Education, Hohhot 010051, China; 3The Inner Mongolia Advanced Materials Engineering Technology Research Center, Hohhot 010051, China

**Keywords:** micro-arc oxidation, zirconium alloy, discharge energy, pulse frequency

## Abstract

The micro-arc oxidation (MAO) technique was used to grow in situ oxidation coating on the surface of R60705 zirconium alloy in Na_2_SiO_3_, Na_2_EDTA, and NaOH electrolytes. The thickness, surface morphology, cross-section morphology, wear resistance, composition, and structure of the micro-arc oxidation coating were analyzed by an eddy current thickness measuring instrument, XPS, XRD, scanning electron microscopy, energy spectrometer, and wear testing machine. The corrosion resistance of the coating was characterized by a polarization curve and electrochemical impedance spectroscopy (EIS). The results show that, with the increase in frequency, the single-pulse discharge energy decreases continuously, and the coating thickness shows a decreasing trend, from the highest value of 152 μm at 400 Hz to the lowest value of 87.5 μm at 1000 Hz. The discharge pore size on the surface of the coating gradually decreases, and the wear resistance and corrosion resistance of the coating first increase and then decrease. The corrosion resistance is the best when the frequency is 400 Hz. At this time, the corrosion potential is −0.215 V, and the corrosion current density is 2.546 × 10^−8^ A·cm^−2^. The micro-arc oxidation coating of zirconium alloy is mainly composed of monoclinic zirconia (m-ZrO_2_) and tetragonal zirconia (t-ZrO_2_), in which the content of monoclinic zirconia is significantly more than that of tetragonal zirconia.

## 1. Introduction

Zirconium alloys are widely used as special structural materials in the nuclear and chemical industries due to their small thermal neutron absorption cross section, excellent high-temperature mechanical properties, and good physical and chemical compatibility with nuclear fuel [1,2,3,4,5,6]. Consequently, high corrosion resistance is a critical requirement for these applications. The service life of zirconium alloys in practical applications largely depends on their wear and corrosion resistance. Surface modification techniques can significantly enhance these properties and thus extend the service life of zirconium alloys [7,8].

Micro-arc oxidation (MAO), also known as plasma electrolytic oxidation (PEO), is a surface modification technique that in situ grows an oxide coating on the surface of metals such as Al, Mg, Ti, and Zr and their alloys. The coatings produced by MAO are firmly bonded to the metal substrate and exhibit excellent wear resistance, corrosion resistance, and electrical insulation properties [9,10,11]. While the MAO technology for aluminum, magnesium, titanium, and their alloys has been extensively studied both domestically and internationally, there is a noticeable paucity of research on the MAO of zirconium alloys.

Several factors influence the MAO coating preparation process, including working voltage, electrolyte composition, oxidation time, frequency, and duty cycle [12,13,14]. Current research has primarily focused on the effects of power supply type, electrolyte composition, voltage, and oxidation time. However, the MAO process is highly complex, involving the interplay of physical, chemical, and electrochemical mechanisms. Pulse frequency, in particular, is a crucial parameter that controls single-pulse energy during the MAO process. Despite its importance, there has been limited research on the impact of single-pulse discharge energy on the properties of MAO ceramic films [15,16,17].

Therefore, this study aims to investigate the effect of single-pulse discharge energy on the properties of MAO coatings on zirconium alloy by adjusting the pulse frequency. By systematically analyzing the influence of frequency on coating thickness, surface morphology, wear resistance, and corrosion resistance, we aim to provide a comprehensive understanding of how single-pulse energy impacts the overall performance of MAO coatings.

## 2. Experimental

### 2.1. Materials and Methods

The matrix material used in the experiment is R60705 zirconium alloy, and its main chemical composition is shown in Table 1.

The zirconium alloy matrix was processed into rectangular specimens measuring 30 mm × 20 mm × 3 mm using wire cutting. The specimens were then degreased, gradually polished with sandpaper, and subjected to ultrasonic cleaning. A complete T-MAO-B30 system was used to prepare the MAO coating, with the electrolyte composition for microarc oxidation consisting of Na_2_SiO_3_ (16.0 g/L) + Na_2_EDTA (2.0 g/L) + NaOH (2.0 g/L), prepared with deionized water. This experiment employed a single-factor analysis method, with the zirconium alloy sample serving as the anode and the electrolytic cell as the cathode. A cooling circulation device was utilized to maintain the electrolyte temperature at around 30 °C. The micro-arc oxidation process was conducted in constant pressure mode, with the process parameters detailed in Table 2.

### 2.2. Characterization

The thickness of the micro-arc oxidation coating of zirconium alloy was measured by an HCC-25 eddy current thickness gauge. The phase analysis of the micro-arc oxidation coating was carried out by a Rigaku SmartLab X-ray diffractometer (the scanning angle was 20~80°; the scanning rate was 2°/min). XPS analysis of the coating was performed by a photoelectron spectrometer. The X-ray source was a monochromatic Al Ka source (hv = 1486.6 eV) with a voltage of 20 kV. The micromorphology of the oxide coating was analyzed by a Hitachi S-3400NII scanning electron microscope. A DZ-322TABER wear testing machine was used to test the wear resistance of the coating (250 g pressure load, 50 r/min speed, 5 times of wear for each sample, and the average value was taken).

X-ray photoelectron spectroscopy was used to analyze the oxide layer with an ESCALAB-250Xi electron spectrometer produced by Thermo Fisher Technology. The X-ray source was a monochromatic Al Ka source (hv = 1486.6 eV) with a voltage of 20 kV.

A Zennium (Zahner, Germany) electrochemical workstation was used to characterize the corrosion properties of the film layer in a 3.5% NaCl solution. The reference electrode was a saturated calomel electrode (SCE), and the auxiliary electrode was a platinum electrode. The sample was wrapped with epoxy resin and protected as a working electrode, with a bare area of 1 cm^2^. Before the experiment, to stabilize the open-circuit potential, the sample was immersed in a NaCl solution for about 1 h. The polarization curve was measured with a scanning range of ±0.6 V and a scanning rate of 1 mV/s. The potential values are given versus the saturated calomel electrode (SCE).

To characterize the corrosion properties of the film layer, electrochemical impedance spectroscopy (EIS) tests were performed on the film layer. The frequency scanning range was from 100 kHz to 1 kHz, and the amplitude of the AC excitation signal was 10 mV. After the experiment, equivalent circuit fitting was performed using Zview (3.2.0.49) software to obtain the resistance and capacitance information of the oxide film.

## 3. Micro-Arc Oxidation Discharge Mechanism and Model

### 3.1. Discharge Mechanism

The micro-arc oxidation coating is formed by the accumulation of metal oxides during the process of micro-arc discharge. The photo-thermal radiation generated by the arc discharge and the heat released by the metal oxidation reaction cause local high temperatures in the discharge area, which result in the generated oxide melting. Under the combined action of the electrolyte and the introduced cooling and circulating water, a thin coating and substrate formed by the initial micro-zone discharge will experience another breakdown effect [18,19,20]. Finally, a metal oxide coating with a non-equilibrium microstructure is formed. In the process of micro-arc oxidation, the pulse energy (*E*) on the surface of the metal sample can be expressed as follows:(1)$$E=\int_{0}^{t}UIdt=\frac{U^{2}D}{Rf}$$
*U*—the voltage of the power supply in the microarc oxidation process;*I*—the magnitude of the positive pulse current;*R*—the equivalent resistance of the pulse-generating discharge circuit;*t*—the discharge time of the power supply during processing;*f*—the frequency of the working power supply;*D*—the pulse duty cycle.

The effective discharge time *t* mainly depends on the frequency *f* of the working power supply and the pulse duty cycle *D*.

### 3.2. Discharge Model

Figure 1 shows the discharge model during micro-arc oxidation of zirconium alloy. The electrochemical reactions that occur during the reaction are as follows:

The main reaction of the anode:4OH^−^ − 4e → 2H_2_O + O_2_↑(2)
Zr − 4e → Zr^4+^(3)
Zr + 2OH^−^ − 2e → ZrO_2_ + H_2_↑(4)

The main reaction of the cathode:2H_2_O + 2e → 2OH^−^ + H_2_↑(5)
O_2_ + 2H_2_O + 4e → 4OH^−^(6)

**Figure 1 materials-17-03166-f001:**
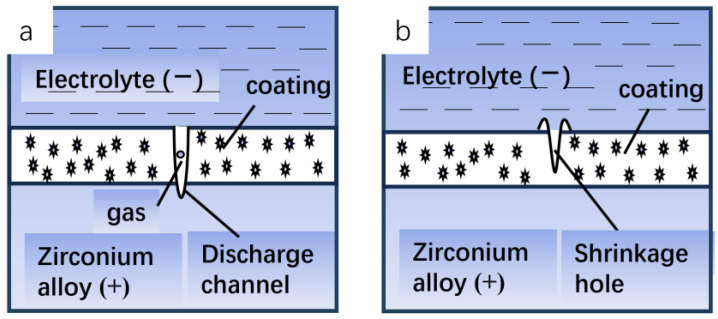
Micro-arc discharge model of zirconium alloy micro-arc oxidation before (**a**) discharge and (**b**) after discharge.

The micro-scale arc discharge model involves the following processes:(1)When the power supply is connected and starts working, due to the small initial voltage, a thin anodic oxidation coating will be formed on the surface of the substrate. At the same time, due to the electrolytic reaction of the electrolyte, a part of the mixed gas with oxygen as the main component will be produced, including a small amount of hydrogen generated by the anodic oxidation reaction, and will be influenced by the composition of the electrolyte.(2)With the increase in oxidation time and the increase in voltage, the coating will continue to thicken, and the precipitated gas will continue to accumulate in the discharge channel. When the voltage reaches the breakdown voltage, the gas will be ionized by the voltage breakdown to form a plasma, which will trigger the discharge effect under the action of the electric field strength. At the same time, denser electric sparks can be observed on the surface of the substrate.(3)In the process of spark discharge in the micro-zone, concentrated energy will be released, causing the temperature in the micro-zone to rise instantly, and the local high temperature will melt the metal matrix to form molten oxide. Due to the discharge pressure generated in the discharge channel and the pressure of gas ejecting outward, the molten oxide with higher temperatures will be ejected outward through the discharge channel. As the coating surface is in contact with the electrolyte, it cools rapidly, the volume shrinks, and a jet vent similar to a crater is formed on the surface of the film layer; and finally, a certain thickness of the oxidation coating is obtained.

## 4. Results and Discussion

### 4.1. Thickness of MAO Coating

Figure 2 shows the variation in coating thickness under different pulse frequencies. It can be seen from the figure that the thickness of the coating does not change significantly when the frequency is between 200 and 600 Hz, maintaining a value of approximately 152 μm. Under low-frequency conditions, the energy per pulse is high, and the pulse duration is sufficient, allowing adequate time for the formation of the oxidation coating. However, with an increase in frequency, the thickness of the micro-arc oxidation coating exhibits a significant decrease, reaching a minimum of 87.5 μm at 1000 Hz.

Pulse frequency refers to the number of effective discharges occurring across the discharge gap per unit time. Therefore, the adjustment of frequency directly affects both the energy and the duration of each pulse. The growth rate of the coating is primarily determined by the discharge energy of the pulse. When other parameters are held constant, increasing the frequency shortens the discharge time of a single pulse, resulting in a continuous reduction in the pulse energy acting on the substrate surface. This leads to a weakening of the breakdown capability and a subsequent reduction in the film formation rate. At high frequencies (1000 Hz), the pulse energy is insufficient to provide adequate energy for the reaction, resulting in a thinner coating thickness.

### 4.2. Coating Micromorphology

Figure 3 shows the surface topography of the MAO coating at different frequencies. The figure illustrates that the coating surface is formed by the accumulation of numerous molten materials resembling “volcanic piles”. During the micro-arc oxidation process, arc discharges occur on the surface of the zirconium alloy sample, resulting in the breakdown of the oxide coating on the substrate. This breakdown leads to the formation of molten oxides and bubbles, which are expelled from the discharge channel and rapidly solidified upon contact with the electrolyte. This rapid solidification and accumulation cause the uneven surface of the coating.

As shown in Figure 3a, when the frequency is 200 Hz, the coating surface is uneven and filled with holes and cracks of varying sizes. As the frequency increases, the energy per pulse decreases, resulting in a gradual reduction in the pore size of the discharge microholes on the coating surface. Consequently, the depressions and cracks on the coating surface are filled, reducing the surface roughness and making the surface smoother and flatter. However, when the frequency is too high, as depicted in Figure 3e, the coating surface becomes gullied. This is because, at high frequency, although the discharge energy is weakened and the reaction rate decreases, gas is still generated during the reaction process. At this point, due to the lower energy, the gas cannot be discharged from the discharge channel in time, leading to the formation of a gullied coating surface.

### 4.3. Coating Phase Composition

Figure 4 shows the X-ray diffraction (XRD) pattern of R60705 zirconium alloy micro-arc oxidation (MAO) coating at different pulse frequencies. The presence of sharp diffraction peaks in the figure indicates that the MAO coating is crystalline. It can be observed from the figure that the phase composition of the MAO coating mainly consists of the monoclinic phase (m-ZrO_2_) and the tetragonal phase (t-ZrO_2_), with the number and intensity of the diffraction peaks for the monoclinic phase being greater than those for the tetragonal phase. Additionally, the Si element from the electrolyte also participates in the reaction, resulting in the presence of a small amount of SiO_2_ in the coating.

As shown in the figure, with the increase in frequency, the intensity of the ZrO_2_ diffraction peak initially increases and then decreases. When combined with the changes in coating thickness, it can be inferred that, at low frequencies, the pulse discharge energy is high, leading to the gradual accumulation of oxides in the coating and, consequently, an increase in the intensity of the ZrO_2_ diffraction peaks. However, as the frequency continues to increase, the pulse discharge energy gradually decreases, and the coating thickness begins to decline. This reduction in oxide content results in a decrease in the intensity of the diffraction peaks.

Figure 5 shows the XPS spectrum and peak distribution of the micro-arc oxidation coating surface. Figure 5a–c present the survey XPS spectrum and the high-resolution spectra of Zr 3d and Si 2p, respectively. From Figure 5a, it can be observed that the composition of the zirconium alloy micro-arc oxidation coating includes the Na, O, Si, C, and Zr elements. Among these, Na and Si originate from Na_2_SiO_3_ in the electrolyte, and Zr originates from the zirconium alloy matrix. The C element may come from the addition of Na2EDTA to the electrolyte and the presence of C in the external environment during the experiment. The binding energy of various elements in the spectrum can be used to characterize the chemical state of elements in the coating.

In Figure 5b, the XPS pattern of Zr on the surface of the sample shows characteristic peaks at 181.38 eV and 183.58 eV, with a peak separation of 2.2 eV. These peaks correspond to Zr 3d_5/2_ and Zr 3d_3/2_, respectively, indicating that zirconium primarily exists in the form of ZrO_2_ in the micro-arc oxidation coating. The characteristic peaks 3d_5/2_ = 181.38 eV and 3d_3/2_ = 183.58 eV correspond to the binding energy of zirconium ions (Zr^4+^). In Figure 5c, the binding energy of Si 2p Is 101.9 eV, corresponding to SiO_2_, indicating that the Si element also participates in the micro-arc oxidation reaction. This is consistent with the presence of a small amount of SiO_2_ in the XRD results.

### 4.4. Coating Wear Resistance

Figure 6 shows the weightlessness of the coating after wear. It can be seen from the figure that the wear quality loss of the coating at different frequencies is roughly the same, and the wear amount is relatively large at the initial wear stage (0~50 rad). This is because, at the beginning of the wear, the outer layer of the coating surface is relatively loose, and because the micro-arc discharge only occurs locally on the surface of the coating at the end, the coating surface forms a micro-convex body with different levels, which is more prone to spalling after contact with the abrasive paper under the external load applied, so the wear quality loss is greater. In the late wear period (100~250 rad), the mass loss is small, and the curve is gradually gentle. It can be clearly seen from the figure that the quality loss of the coating reaches the maximum, and the wear resistance is the worst when the frequency is 200 Hz. This is because the pulse discharge energy is large at this time, the reaction rate is strengthened, and the temperature is rapidly increased. A large number of molten oxides generated during the reaction process are ejected from the discharge channel, chilled in contact with the electrolyte, rapidly solidified, and deposited on the surface of the coating, resulting in loose and porous coating, poor density, low hardness, and the easiness of being affected by wear. With the increase in frequency, the coating loss decreases and the wear resistance increases. This is because, with the increase in frequency, the monopulse discharge energy decreases, resulting in the plasma discharge tending to moderate, so that the condensation products after each plasma discharge are reduced, the surface roughness is reduced, and the surface tends to be smooth and flat.

Figure 7 shows the micro morphologies of MAO coatings at different frequencies. It can be seen from Figure 7a that, after the coating is subjected to wear testing, the wear marks on the surface of the coating are deep and widely distributed, indicating severe damage. As derived from Equation (1), when other parameters are constant, a lower frequency results in greater discharge energy on the sample surface. This leads to an increase in the oxide released by the discharge micromoles, which easily gushes out of the pores and sprays onto the coating surface. Subsequently, the melt rapidly cools upon contact with the electrolyte and is deposited at the passageway near the surface of the coating, forming a relatively loose and irregular convex film surface. This increases the surface roughness and makes it more prone to spalling when in contact with sandpaper.

As shown in Figure 7b,c, the worn surfaces are relatively intact, and the wear marks become shallower. As the pulse frequency increases, the pulse energy weakens and the intensity of the reaction decreases. The oxides released by the discharge micropores begin to accumulate inside the coating, which improves the coating stability.

### 4.5. Corrosion Resistance of Coating

The polarization curves and fitting parameters of the zirconium alloy micro-arc oxidation films immersed in a 3.5 wt% NaCl solution at various frequencies are presented in Figure 8 and Table 3, respectively. Ecoor represents the self-corrosion potential, indicating the material’s susceptibility to corrosion, while Jcorr denotes the corrosion current density, reflecting the corrosion rate of the metal.

Table 3 illustrates that the micro-arc oxidation coating exhibits the highest corrosion potential and corrosion current density at 200 Hz, indicating greater susceptibility to corrosion and a faster corrosion rate. At 400 Hz, the corrosion potential shifts positively from −0.276 V to −0.215 V. Correspondingly, the corrosion current density decreases from 1.018 × 10^−6^ A·cm^−2^ to 2.546 × 10^−8^ A·cm^−2^, which is two orders of magnitude lower than that at 200 Hz. This suggests that increasing the frequency enhances the coating’s corrosion resistance, effectively impeding the penetration of corrosive media into the coating and improving the material’s overall corrosion resistance.

The coating exhibits a relatively flat surface with fewer cracks at 400 Hz, resulting in the smallest corrosion current density and the highest corrosion resistance among the tested frequencies. However, as the frequency increases further (600 to 1000 Hz), the self-corrosion current density initially increases and then decreases, although it remains higher than that observed at 400 Hz. Consequently, the corrosion resistance at these frequencies is inferior to that observed at 400 Hz. Therefore, it can be concluded that the micro-arc oxidation coating demonstrates optimal corrosion resistance at a frequency of 400 Hz.

To further investigate the corrosion resistance of the micro-arc oxidation coating on R60705 zirconium alloy, coatings prepared at different duty cycles were immersed in a 3.5% NaCl solution for 30 min, followed by electrochemical impedance spectroscopy (EIS) testing using an electrochemical workstation. The experimental results are presented in Figure 9. As depicted in a Nyquist plot in Figure 9a, the arc radius of the bulk resistance reflects the corrosion resistance of the micro-arc oxidation coating. In the Nyquist diagram, the arc radius of the capacitive reactance is directly proportional to the corrosion resistance; thus, a larger arc radius indicates a higher corrosion resistance.

Figure 9a shows that, at a frequency of 400 Hz, the arc radius of the capacitive reactance is the largest, indicating the highest corrosion resistance. As the frequency increases beyond this point, the corrosion resistance decreases. This phenomenon can be attributed to the moderate discharge energy at 400 Hz, allowing sufficient time for the formation of a dense oxide layer. This slow deposition rate promotes coating density and uniformity, thereby enhancing the corrosion resistance. In contrast, higher frequencies result in reduced discharge energy, preventing the formation of a dense oxide layer within the coating, consequently diminishing its corrosion resistance.

The impedance modulus in the Bode plot is denoted as |Z|, which also serves as an indicator of corrosion resistance. As depicted in Figure 9b, zirconium alloy exhibits optimal corrosion resistance at a frequency of 400 Hz, with the corrosion resistance decreasing sequentially at frequencies of 800 Hz, 1000 Hz, 600 Hz, and 200 Hz. These findings align with the trends observed in the polarization curves and Nyquist diagrams.

The frequency variation influences the rate of oxide formation, deposition rate, and internal structure during the micro-arc oxidation process. Therefore, selecting appropriate process parameters is crucial in optimizing the density and uniformity of the coating. This approach ultimately enhances the corrosion resistance of the micro-arc oxidation coating.

Table 4 presents the fitting data of the zirconium alloy micro-arc oxidation coating obtained from electrochemical impedance spectroscopy (EIS) at various frequencies, while Figure 10 illustrates the corresponding fitting circuit diagram of EIS. In the diagram, Rs represents the solution resistance between the element and the reference electrode. Rc and CPEc denote the resistance and capacitance of the outer loose layer, respectively. CPEdl and Rct represent the double-layer capacitance and interfacial charge transfer resistance between the solution and the film, with the polarization resistance Rp calculated as Rp = Rc + Rct.

It is evident from the table that as the frequency increases, both Rc and Rct of the coating exhibit a trend of initial increase, followed by a decrease, reaching their peak values at 400 Hz. At this frequency, Rp attains a value of 1.17 × 10^4^ Ω·cm^2^, indicating the highest resistance. Greater polarization resistance implies lower current density through the coating, indicating a stronger corrosion resistance. However, as the frequency further increases, Rp begins to decrease.

Observing Table 4 reveals that the polarization resistance of the coating varies in the order of 400 Hz > 800 Hz > 1000 Hz > 600 Hz > 200 Hz, from highest to lowest. These results are consistent with those obtained from polarization curves, Nyquist diagrams, and Bode plots.

## 5. Conclusions

(1)The low frequency has little effect on the thickness of the coating. With the increase in frequency, the monopulse discharge energy continues to decrease, and the coating thickness decreases significantly, and it reaches the lowest level at 1000 Hz. With the increase in frequency, the pore size of the discharge on the surface of the coating decreases and the surface tends to be smooth and flat.(2)The coating is mainly composed of the monoclinic phase (m-ZrO_2_), tetragonal phase (t-ZrO_2_), and SiO_2_ phase. The binding energies of Zr are 181.38 eV (3d_3/2_) and 183.58 eV (3d_5/2_). The oxidation coating is mainly composed of the Zr, O, and Si elements.(3)With the increase in frequency, the wear resistance of oxide coating first increases and then decreases. The corrosion resistance increases first and then decreases. The corrosion resistance is best at 400 Hz, when the corrosion potential is −0.215 V and the corrosion current density is 2.546 × 10^−8^ A·cm^−2^.

## Figures and Tables

**Figure 2 materials-17-03166-f002:**
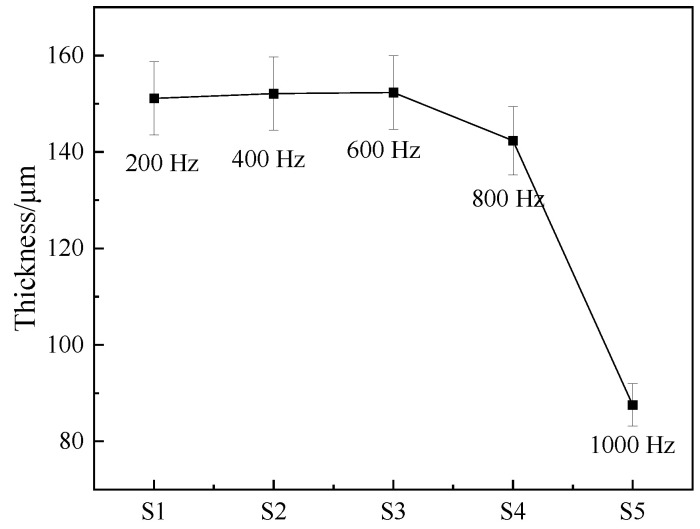
Layer thickness at different frequencies.

**Figure 3 materials-17-03166-f003:**
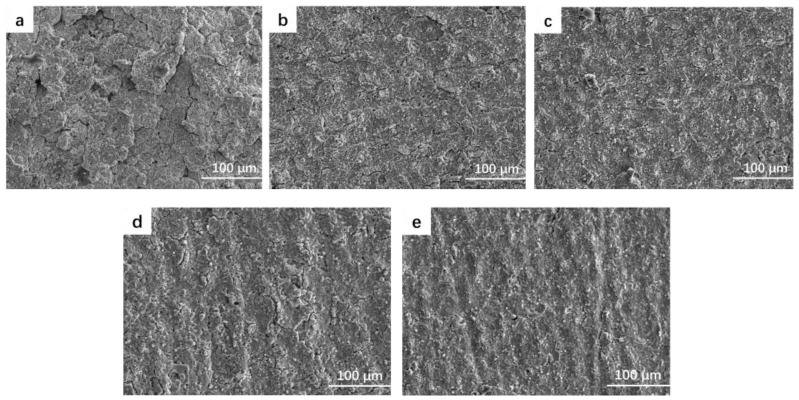
Surface micromorphology of the oxide film: (**a**) S1, (**b**) S2, (**c**) S3, (**d**) S4, and (**e**) S5.

**Figure 4 materials-17-03166-f004:**
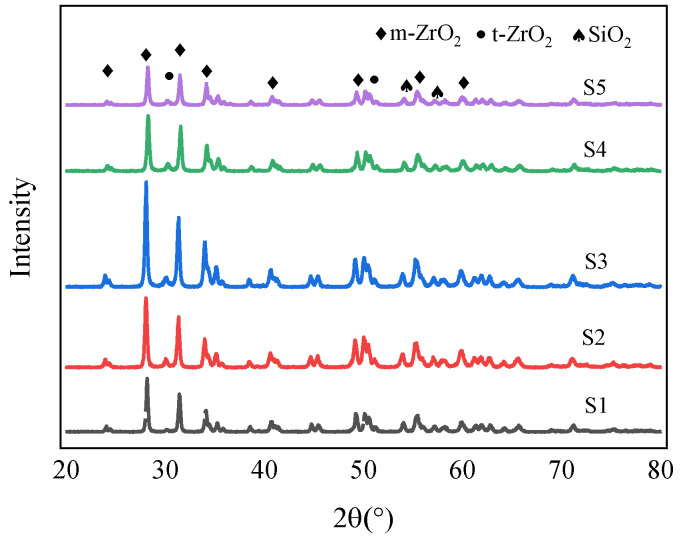
X-ray diffraction patterns of an oxide film at different frequencies.

**Figure 5 materials-17-03166-f005:**
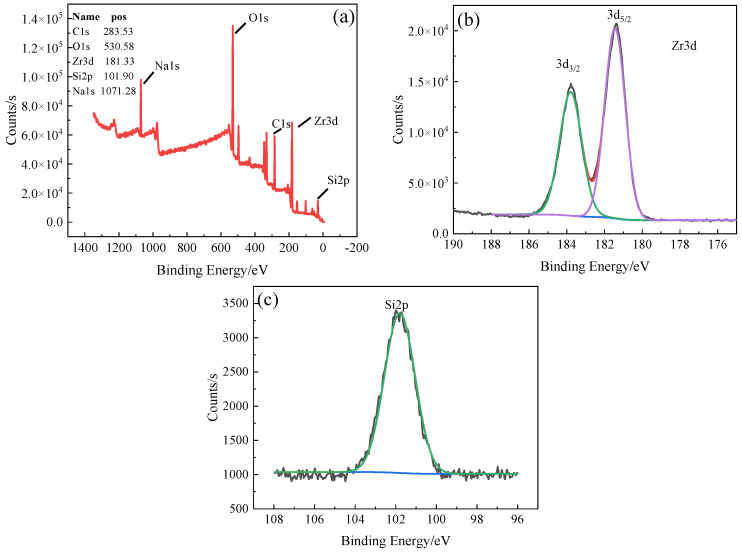
XPS analysis of MAO coating: (**a**) general spectrum, (**b**) peak plot of Zr3d, and (**c**) peak plot of Si2p.

**Figure 6 materials-17-03166-f006:**
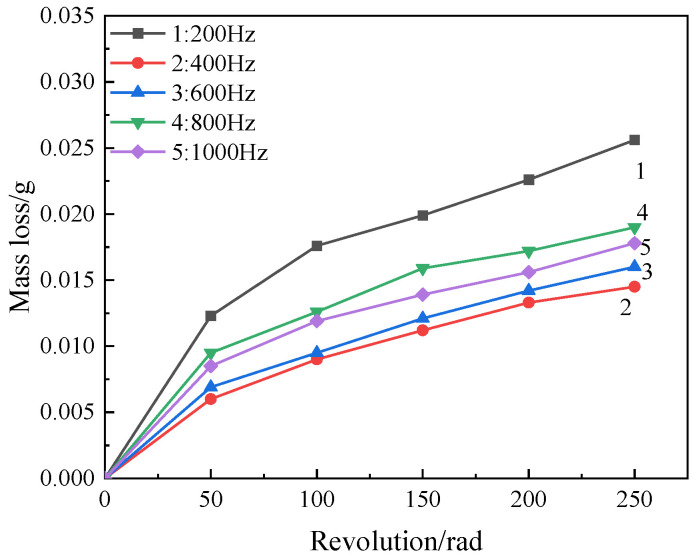
Mass loss of the coating at different frequencies.

**Figure 7 materials-17-03166-f007:**
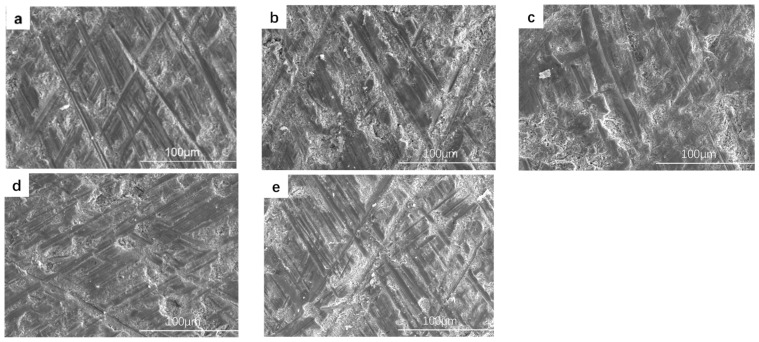
The wear morphology of the coating at different frequencies: (**a**) S1, (**b**) S2, (**c**) S3, (**d**) S4, and (**e**) S5.

**Figure 8 materials-17-03166-f008:**
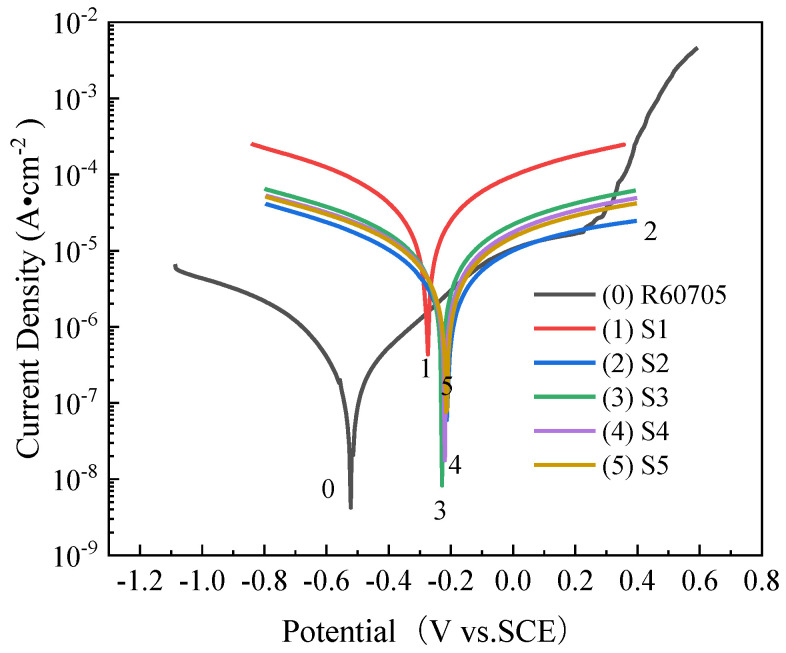
Polarization curves of micro-arc oxide film at different frequencies.

**Figure 9 materials-17-03166-f009:**
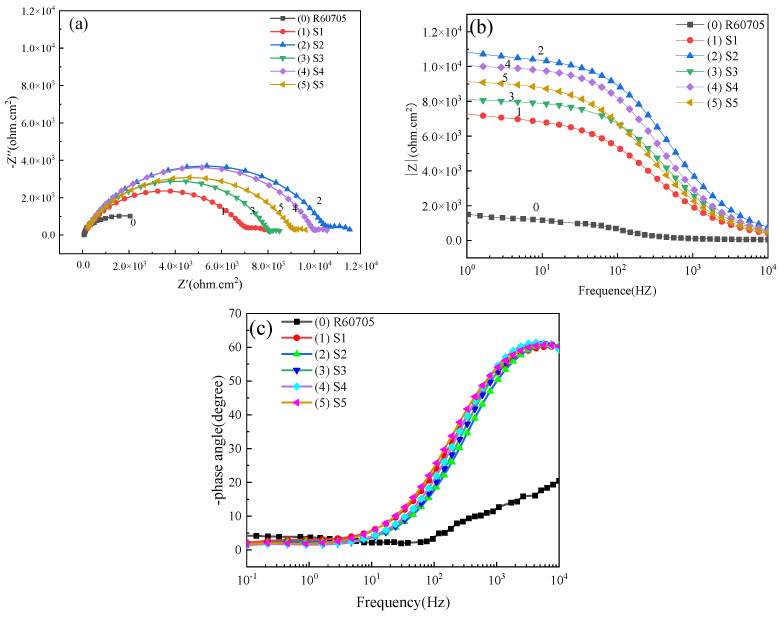
Nyquist and Bode patterns of micro-arc oxide films at different frequencies: (**a**) Nyquist plots, (**b**) Bode plots of |Z| vs. frequency, and (**c**) Bode plots of phase angle vs. frequency.

**Figure 10 materials-17-03166-f010:**
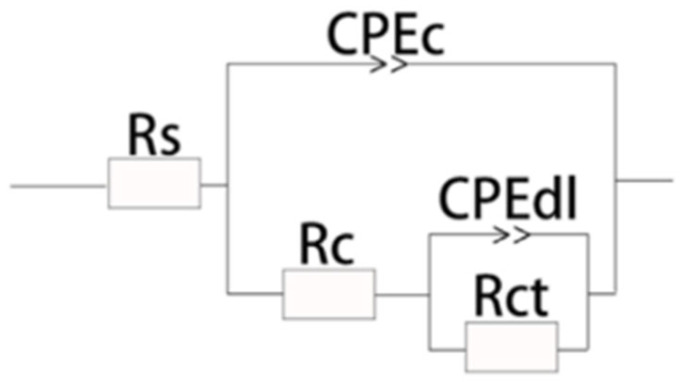
MAO coating equivalent circuit diagram.

**Table 1 materials-17-03166-t001:** Chemical composition of zirconium alloy (wt.%).

Element	Hf	Nb	Fe + Cr	C	N	H	O	Zr
**Content%**	4.5	2~3	0.2	0.05	0.025	0.005	0.18	margin

**Table 2 materials-17-03166-t002:** Process parameters in the micro-arc oxidation process.

Specimen Code	Frequency (Hz)	Forward/Negative Voltage (V)	Duty Cycle (%)	Oxidation Time (min)
S1	200	350/140	50	15
S2	400			
S3	600			
S4	800			
S5	1000			

**Table 3 materials-17-03166-t003:** Fitting parameters of polarization curves.

Frequency/Hz	E_coor_/V vs. SCE	J_corr_/(A·cm^−2^)
R60705	−0.529	5.60 × 10^−6^
S1	−0.276	1.02 × 10^−6^
S2	−0.215	2.55 × 10^−8^
S3	−0.228	6.23 × 10^−8^
S4	−0.220	3.39 × 10^−8^
S5	−0.215	4.81 × 10^−8^

**Table 4 materials-17-03166-t004:** Coating EIS fitting results.

Frequency/Hz	Rs (Ω·cm^2^)	Rc (Ω·cm^2^)	Rct (Ω·cm^2^)	Rp (Ω·cm^2^)
R60705	43.65	-	976	9.76 × 10^2^
S1	39	7002	1105	8.11 × 10^3^
S2	91.38	10,448	1212	1.17 × 10^4^
S3	85.6	7958	669.2	8.63 × 10^3^
S4	91.3	9892	743.7	1.06 × 10^4^
S5	47.6	9102	685.7	9.79 × 10^3^

## Data Availability

Data are contained within the article.

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
