# Peer review of "Effect of Discharge Energy on Micro-Arc Oxidation Coating of Zirconium Alloy"

_materials, 2024, doi:10.3390/ma17133166_

Round 1

Reviewer 1 Report

Comments and Suggestions for Authors

The paper "Effect of discharge energy on micro arc oxidation coating of zirconium alloy" considers an interesting subject and should be definitely of interest for the readers of Metals. However, the following corrections are recommended to be done:

1) To write the Introduction more developed and thus to add more references to it.

2) To check thoroughly the language of the paper.

3) To edit the references, because their formatting is very poor.

Comments on the Quality of English Language

It is necessary to check thoroughly the language of the paper.

Author Response

Comments 1: To write the Introduction more developed and thus to add more references to it.

Response 1: Thank you for your review and valuable comments on my manuscript. Based on your suggestions, I have expanded the introduction and added more references to provide a more comprehensive research background and related work.

Comments 2: To check thoroughly the language of the paper.

Response 2: Thank you for your suggestion. I will carefully check the language of the paper to ensure its accuracy and fluency. Thank you very much for your patience and guidance.

Comments 3: To edit the references, because their formatting is very poor.

Response 3: Thank you for your review and valuable comments on my manuscript. I will immediately carefully edit and adjust the format of the reference in accordance with the journal's requirements and format guidelines to ensure that it meets the standards.

Additional clarifications

Parts of the text that have been revised have been highlighted in yellow

Reviewer 2 Report

Comments and Suggestions for Authors

In the manuscript, the authors studied the effect of monopulse discharge energy on the properties of zirconium alloy micro-arc oxidation coating by adjusting the monopulse frequency. 

Zirconium alloy is widely used as a special structural material in the nuclear industry and chemical industry, so the topic may be of interest to scientific community and I believe the paper is worth publishing.

The manuscript is well organized and well written. Abstract briefly summarizes the findings of the research. Introduction contains a description of the reasons for choosing the topics of the work. Materials, the process parameters in the micro-arc oxidation process, and methods were sufficiently described. The authors also described the microarc oxidation discharge mechanism and model. The research results are correctly discussed and explained to the reader. Figures and graphic material are good quality. Conclusions briefly summarize the research results obtained. References contain 16 items, all dated 2018-24.

Taking into account the above-mentioned advantages of the manuscript, I recommend accepting the article for publication.

Before publishing, I suggest minor revision for the improving the quality of the manuscript:

1) section Characterization, line 77 
There is no information about the methodology of the electrochemical impedance spectroscopy: What metal was the auxiliary electrode made of? What was the amplitude of the AC perturbation signall? 

2) section Characterization, line 78 
There is no information regarding the methodology for measuring polarization curves: What metal was the auxiliary electrode made of? What reference electrode was used? What was the rate of change of potential?

3) Table 3, column: Ecoor 
Please add the type of reference electrode, versus to which the potential values are given.

4) Figure 9a) - Nyquist plots 
It is commonly accepted that the scale is the same on the Z' and Z" axes. Spaces between markers should be the same on both axes and length of the axes should be proportional to each other. I recommend rearranging the plots keeping the above-mentioned rule. Then there will be semicircles in the plots visible, not "semi-ellipses", which will make it easier to interpret the results.

Author Response

Comments 1: section Characterization, line 77

There is no information about the methodology of the electrochemical impedance spectroscopy: What metal was the auxiliary electrode made of? What was the amplitude of the AC perturbation signall?

Response 1: Thanks for your guidance, I have added information on the auxiliary electrode material and AC disturbance signal amplitude in the methodology section. The auxiliary electrode used is a platinum electrode, and the amplitude of the AC perturbation signal is 10 mV.

Comments 2: section Characterization, line 78

There is no information regarding the methodology for measuring polarization curves: What metal was the auxiliary electrode made of? What reference electrode was used? What was the rate of change of potential?

Response 2: I have added the information about the auxiliary electrode and reference electrode materials in the methodology section. The auxiliary electrode used is a platinum electrode, and the reference electrode is a saturated calomel electrode (SCE). The rate of change of potential is 1 mV/s.

Comments 3: Table 3, column: Ecoor

Please add the type of reference electrode, versus to which the potential values are given.

Response 3: Thank you for your review and valuable comments on my manuscript. I specify the type of reference electrode in the methodology section and point out that all potential values are given relative to the saturated calomel electrode (SCE).

Comments 4:Figure 9a) - Nyquist plots 
It is commonly accepted that the scale is the same on the Z' and Z" axes. Spaces between markers should be the same on both axes and length of the axes should be proportional to each other. I recommend rearranging the plots keeping the above-mentioned rule. Then there will be semicircles in the plots visible, not "semi-ellipses", which will make it easier to interpret the results.

Response 4:

Additional clarifications

Parts of the text that have been revised have been highlighted in yellow

Reviewer 3 Report

Comments and Suggestions for Authors

The authors's report presents important experimental results obtained by testing the most important materials properties (chemical, structural, mechanical, corrosion) of the surface oxide layers developed by the still relatively new micro-arc oxidation (MAO) technique via converting specimens made of a zirconium alloy.

The paper is concise (not too long) and concentrates on the experimental results with also presenting rational and logical qualitative explanations to them in a quite sound way.

The mechanism of the discharge conversion process is explained in a well composed and easily understandable

Probably the letter size in Eq.(1.1) could be modified to the same size; moreover the frequencies as well could be presented in Figure 2. In Table 1 there are some misprints like "elment" and "Cotent%". In Line72 probably the letter for the frequency is not v but the gregg letter nu. In Line 213 I can not fully understand the phrase "wear quality loss rule". Is not it "quantity"? Also in Figure 6 correct the word "Mass lose" to "Mass loss". In Figure 7. in not something missing from the sentence? In Line 130 I would write "As the coating..." instead of "After..." Also, please correct the subscripts in Table 3   (corr). 

Moreover, the list of References contains a lot of obscure capital letters/initials (instead of names), etc. so this part of the paper should be thoroughly checked and corrected, please.

Otherwise, the manuscript methodology applied as well as its presentation is of high quality and contains much new and useful information. 

Author Response

Comments 1: Probably the letter size in Eq.(1.1) could be modified to the same size; moreover the frequencies as well could be presented in Figure 2. In Table 1 there are some misprints like "elment" and "Cotent%". In Line72 probably the letter for the frequency is not v but the gregg letter nu. In Line 213 I can not fully understand the phrase "wear quality loss rule". Is not it "quantity"? Also in Figure 6 correct the word "Mass lose" to "Mass loss". In Figure 7. in not something missing from the sentence? In Line 130 I would write "As the coating..." instead of "After..." Also, please correct the subscripts in Table 3 (corr).

Response 1: Thank you for your suggestions. I have corrected all the questions you raised in the original text.

Comments 2: Moreover, the list of References contains a lot of obscure capital letters/initials (instead of names), etc. so this part of the paper should be thoroughly checked and corrected, please.

Response 2: Thank you for your review and valuable comments on my manuscript. I will immediately carefully edit and adjust the format of the reference in accordance with the journal's requirements and format guidelines to ensure that it meets the standards.

Comments 3: Otherwise, the manuscript methodology applied as well as its presentation is of high quality and contains much new and useful information.

Response 3: Thank you very much for your patience and guidance. Please feel free to let us know if you have any further suggestions or comments.

Additional clarifications

Parts of the text that have been revised have been highlighted in yellow